# Researching the hard-to-reach: a scoping review protocol of digital health research in hidden, marginal and excluded populations

Rachel Victoria Belt [1], Kazem Rahimi [1], Samuel Cai [2]

¹Nuffield Department of Women's & Reproductive Health, University of Oxford, Oxford, UK
²Centre for Environmental Health and Sustainability, Department of Health Sciences, University of Leicester, Leicester, UK

**Correspondence to**
Rachel Victoria Belt;
rachel.belt@wrh.ox.ac.uk

## ABSTRACT

**Introduction** There is a significant growth in the use of digital technology and methods in health-related research, further driven by the COVID-19 pandemic. This has offered a potential to apply digital health research in hidden, marginalised and excluded populations who are traditionally not easily reached due to economic, societal and legal barriers. To better inform future digital health studies of these vulnerable populations, we proposed a scoping review to comprehensively map published evidence and guidelines on the applications and challenges of digital health research methods to hard-to-reach communities.

**Methods and analysis** This review will follow the Arksey and O' Malley methodological framework for scoping reviews. The framework for the review will employ updated methods developed by the Joanna Briggs Institute including the Preferred Reporting Items for Systematic reviews and Meta-Analysis Scoping Review checklist. PubMed, the Cochrane Library, PsycINFO, Google Scholar and Greenfile are the identified databases for peer-reviewed quantitative and qualitative studies in-scope of the review. Grey literature focused on guidance and best practice in digital health research, and hard-to-reach populations will also be searched following published protocols. The review will focus on literature published between 1 February 2012 and 1 February 2022. Two reviewers are engaged in the review. After screening the title and abstract to determine the eligibility of each article, a thorough full-text review of eligible articles will be conducted using a data extraction framework. Key extracted information will be mapped in tabular and visualised summaries to categorise the breadth of literature and identify key digital methods, including their limitations and potential, for use in hard-to-reach populations.

**Ethics and dissemination** This scoping review does not require ethical approval. The results of the scoping review will consist of peer-reviewed publications, presentations and knowledge mobilisation activities including a lay summary posted via social media channels and production of a policy brief.

## STRENGTHS AND LIMITATIONS OF THIS STUDY

⇒ This review will be the first which considers the intersection of digital health research and marginalised, hidden or excluded populations without a narrow focus on a platform or specific population.

⇒ The rapid expansion in the use of digital health research risks further exacerbation of the digital divide among populations. This review will provide timely information on targeted strategies to promote and improve inclusivity of hard-to-reach populations in digital health research.

⇒ The inclusion criteria for this scoping review is set to be broad, including any digital research on both the environmental and social determinants of health in hard-to-reach populations. This allows a thorough review of existing digital methods used across different sectors and disciplines that may have a wide applicability to hard-to-reach population research.

⇒ The synthesis of data will be limited to a mapping of methods and categorisation and less on the strength of each study due to the variance across the different fields of study and specialisations.

⇒ The search is limited in publications in English language only.

populations and subpopulations. Marginalised, hidden and underserved populations are subject to barriers which affect the achievement of equitable health and social welfare outcomes. Populations residing in informal settlements, for example, face many socioeconomic, environmental and legal barriers that prohibit their access to health service. For example, they may lack formal government recognition; therefore, their health status and needs are not tracked sufficiently by official systems. This poor data presents a formidable challenge for both researchers and policymakers to redress health inequities among these populations.[1]

Digital research, defined as 'the use of online and digital technologies to collect and analyse research data', was increasing prepandemic.[2] COVID-19 has driven a digital

## INTRODUCTION

Globally, health equity is not consistently or systematically applied within or across

transformation in healthcare, with increases in access to telehealth, the use of big data for monitoring the pandemic and the deployment of health communications via social media and other digital platforms.[3–5] The use of online surveys, mobile devices and social media are examples of digital modalities used in digital research and to study the impact and spread of COVID-19.[3 6 7] The speed and breadth of COVID-19 research was impressive, yet commentaries like those of Jung *et al* raise concerns regarding the adherence to research quality and standards in terms of study design, data quality and analytical methods.[8] The latest guidance in the area of digital health such as the WHO Digital Health Guidelines and the Digital Health Equity Framework are opportunities to assess the quality of digital research.[9 10]

An improved understanding of the health for hard-to-reach populations is required to achieve the Sustainable Development Goals and to end the COVID-19 pandemic, which has displayed health and social disparities existing within and between countries.[11–13] For this scoping review, hard-to-reach populations are defined as populations who are underserved by health and social services and/or face barriers in achieving equitable health and well-being due to socioeconomic determinants of health, discrimination, location, disability or exposure to climate and environmental factors. Without further review of research methods at the intersection of hard-to-reach populations and digital research, and clarity on barriers to research inclusion, there is a risk that these populations will be further excluded, and health disparities, partly due to the increasing digital divide, will be exacerbated.[14 15]

Application of digital research methods represents an opportunity to improve health equity and the inclusion of traditionally hard-to-reach, underserved and marginalised populations in research to inform programme design and policy. Previous reviews have documented strategies to improve research with socially disadvantaged groups and reviewed specific digital platforms such as social media and the use of specific health research technologies. However, none have systematically documented the methods for applying digital research methods to a hard-to-reach population.[3 16–24]

This review will build on the previous reviews including by Bonevski and Shaghaghi,[22 23] which systematically documented barriers and strategies for research with socially disadvantaged and vulnerable groups. Our scoping review has considered their findings in the design, but we will narrow the review to studies in which research was conducted, digitally. O'Conner,[16] Helena,[17] Adjekum,[19] Giustini,[18] Whitaker[25] and Edo-Osagie[21] published scoping, literature or systematic reviews, or frameworks and analysis of digital health interventions, social media in public health and digital health app development standards. However, these reviews are broader than the scopes in our proposed review, in which we will focus on the methods' application to hard-to-reach populations and the specific barriers or concerns when researching the hard-to-reach.[16–19 21 25] A retrospective

analysis of National Institute of Health funded digital health research by Nebeker documented the increased application of social media platforms in the study of hard-to-reach populations. The analysis concluded that guidance to researchers is insufficient in evaluating key concerns around the use of social media in research.[26]

The objective of this scoping review is to identify appropriate digital research methods to improve health research inclusivity, documenting both strategies and limitations of digital research methods with hard-to-reach populations. This review is important to ensure that the growth in digital methods does not leave behind populations of concern due to design limitations. The timing of this review is critical due to the risks of the COVID-19 pandemic on already vulnerable populations and the need to reach and include the most vulnerable and marginalised in research across health and social disciplines to improve health equity.

## METHODS AND ANALYSIS
A scoping review was chosen due to the broad nature of the subject. In line with the methods developed by Arksey and O' Malley, this review will follow the framework for conducting a scoping study; stage 1: identifying the research question, stage 2: identifying relevant studies, stage 3: study selection, stage 4: charting the data and stage 5: collating, summarising and reporting the results.[27] Detail on each stage is provided.

### Stage 1: identifying the research question
The key research questions outlined in box 1 provide focus to the aim of the review to improve understanding of how digital research methods are applied to hard-to-reach populations including identification of methods, potential challenges and ethical concerns. To inform the search for marginalised, excluded and hidden populations which may not be defined by a term, a list of populations of interest have been included in data extraction framework in online supplemental annex A.

### Stage 2: identifying relevant studies
The search strategy aims to be comprehensive in its inclusion of studies across different socioeconomic and environmental determinants of health among hard-to-reach

---

**Box 1  Research questions**

1. How are digital health research methods applied to marginalised, excluded and hidden populations?

The secondary research questions focus on the strengths and limitations of the identified methods which include:

1. What are digital research methods commonly applied to include or target hard-to-reach populations?
2. What are challenges in recruitment, retention and response rate in digitally designed studies for hard-to-reach populations?
3. What were identified areas of ethical, safety and anonymity concerns in using digital research platforms for hard-to-reach populations?

---

**Table 1** Databases and sources

| Database/source | Focus | Article limit per search term combination |
|---|---|---|
| Published literature | | |
| PubMed | Health and medical research | None |
| Cochrane Library | Relevant systematic reviews on digital health | None |
| PsycINFO | Mental health and social research | None |
| Google Scholar | All search terms | 200 |
| Greenfile | Impact of climate change on health | None |
| Grey literature | | |
| Key technical agencies (WHO, UNICEF, etc) | Hard-to-reach/marginalisation population specific research guidance—digital and non-digital | 50 |
| Campbell Collaboration | Social sciences | 50 |

populations. The example evidence map is available in online supplemental annex B. The selection of health, social science, mental health and climate/environmental research databases will be employed in this review.[28] The database, description and article limit can be found in table 1.

While we primarily focus on peer-reviewed studies for this scoping review, we will also search grey literature, technical reports and policy briefs produced by key agencies following a novel approach as set out by Enticott *et al* (eg, the WHO, UNICEF and the Campbell Collaboration).[29] These reports, guidance and sector best practice will complement our primary evidence mapping based on individual studies to ensure the most complete view of evidence is achieved. The search strategy is available in the online supplemental file.

A population, concept and context framework for this review is illustrated in table 2.[30] Various search terms will be included, namely common hard-to-reach populations, socioeconomic and environmental determinants of health, and names of various digital platforms and methods. These are in line with the draft data extraction framework. Completed search terms will be documented and shared in the final publication.

### Stage 3: study selection
The review has established wide inclusion criteria. The geographical scope of this research is global, but studies are limited to those published in English. Inclusion and exclusion criteria can be found in table 3. Rayyan and Endnote will be used to store all search results. The first reviewer will screen all titles and abstracts against the criteria. The second reviewer will screen 10% of these to validate the screening process for inclusion and exclusion criteria. The first reviewer will then import the full text of all included documents for detailed analysis.

### Stage 4: charting the data
The data charting will occur using a google form collecting the recommended general information about the study according to the guidance for scoping reviews and expanding data collection through a data extraction framework into key study characteristics, including type of digital study, methodology and limitations. The data extraction framework, available in online supplemental annex C, will be tested by both reviewers for accuracy and applicability and applied consistently across all included studies. The first reviewer (RVB) will extract all data, and the second reviewer (SC) will review 10% of the articles to check for alignment. If misalignment is identified, the second reviewer will provide a full review of the literature. The grey literature search will be performed by the first reviewer.

### Stage 5: collating, summarising and reporting the results
The data collected in the charting exercise will be analysed by both reviewers. The digital methods deployed will be documented with particular focus to how the method enabled inclusivity of the hard-to-reach population and followed ethical guidance for research. The limitations of the studies will be analysed with a focus on research strength, ethics, replicability, community engagement and feasibility. Quality of the reviewed studies will be assessed using checklists informed from Joanna Briggs Institute (https://jbi.global/critical-appraisal-tools) for different types of studies. Additional analysis will include the identification of key research gaps and future research priorities at the intersection of digital health research and marginalised populations.

**Table 2** Population, concept and context (PCC) framework

| PCC | Definition | Example |
|---|---|---|
| Population | Hard-to-reach | Homeless |
| Concept | Digital research methods | Online survey |
| Context | Health and social research | Mental health |

### PATIENT AND PUBLIC INVOLVEMENT
No patient involved.

 

**Table 3** Inclusion and exclusion criteria

| Theme | Inclusion criteria | Exclusion criteria |
| --- | --- | --- |
| Geographic | All | None |
| Time | 1 February 2012 to 1 February 2022 | Before 2012 |
| Language | English | All other languages |
| Populations | Populations of interests documented as keywords in the search strategy and the data extraction framework (available in online supplemental annex C) | Digital health studies which are not focused on hard-to-reach/marginalised populations |
| Type of literature | Published, peer-reviewed literature Grey literature including technical reports and policy briefs that are endorsed by key agencies | Literature which is not peer-reviewed Unpublished literature which is not considered sector guidance including presentations, webinars and case studies |
| Digital application | Digital research approaches including digital data collection, use of apps to collect data, digital surveys, use of social media and other strategies, including using machine-learning analytical techniques | Digital interventions such as remote consultations, digital health communications, social media health campaigns Digital information systems which are not part of a research design method will be excluded |

## ETHICS AND DISSEMINATION

This scoping review does not require ethical approval as it reviews already available publications. The results of the scoping review will consist of peer-reviewed publications, presentations and knowledge mobilisation activities including a lay summary posted via social media channels and production of a policy brief.

**Contributors** RVB: conceived the initial idea, designed the review and drafted the manuscript. KR and SC: provided inputs to the study protocol and revised manuscript. All authors approved the manuscript for submission.

**Funding** This research was funded by the PEAK Urban programme, UKRI's Global Challenge Research Fund, Grant Ref: ES/P011055/1; supported by the National Institute for Health Research (NIHR) Oxford Biomedical Research Centre (BRC), the Oxford Martin school, University of Oxford and the British Heart Foundation grant ref: PG/18/65/33872. YSC acknowledges support from National Institute for Health Research (NIHR) Health Protection Research Unit in Environmental Exposures and Health, a partnership between the UK Health Security Agency, the Health and Safety Executive and the University of Leicester. The views expressed are those of the author(s) and not necessarily those of the NHS, the NIHR, the Department of Health and Social Care or the UK Health Security Agency.

**Competing interests** None declared.

**Patient and public involvement** Patients and/or the public were not involved in the design, or conduct, or reporting, or dissemination plans of this research.

**Patient consent for publication** Not applicable.

**Provenance and peer review** Not commissioned; externally peer reviewed.

**ORCID iDs**
Rachel Victoria Belt http://orcid.org/0000-0002-0078-1236
Kazem Rahimi http://orcid.org/0000-0002-4807-4610
Samuel Cai http://orcid.org/0000-0003-1601-2199

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
