## [Reviewer comments · BMJ Open]

ARTICLE DETAILS

TITLE (PROVISIONAL)	Researching the hard-to-reach: a scoping review protocol of digital health research in hidden, marginal, and excluded populations
AUTHORS	Belt, Rachel; Rahimi, Kazem; Cai, Samuel

VERSION 1 – REVIEW

REVIEWER	Chaitali Sinha International Development Research Centre, Global Health
REVIEW RETURNED	28-Mar-2022

GENERAL COMMENTS	This paper is addressing a very timely and important issue. Please find below the concerns I have with it: 1) Given the vast nature of the paper, encompassing several populations and the breadth of digital health research methods and approaches, the details and analysis in the scoping review are not commensurate to what is needed for this type of contribution.2) Looking at the summary of strengths and limitations, I note that both the second and the third bullet are quite broad and rather unclear. For instance, it seems like the second bullet is about inequalities and inequities exacerbated by COVID-19, which are subsequently overlaid with other inequities in health and access to digital tools. But, this is a bit of a guess and the current wording makes me unsure what is being communicated. General comment: there seems to be too many ideas being conveyed in this bullet. - the third bullet in this list also needs to be tightened up. Is the applicability wide because of the SDOH framing or because the inclusion criteria for the digital health research is not disease or population-specific?- I find some of the general statements like the first one in the paper a bit misleading. Currently it states: "Health equity is a common term in policy and research, although significant barriers remain towards achieving health for all." - I don't think the fact that a certain term is common in policy and research should translate into it being overcome. I would nuance this statement. Perhaps by saying that health equity is not applied consistently or systematically within or across populations and sub-populations.- For the framing of the paper, it might be useful to state that marginalized, hidden and underserved populations are difficult to serve and redress health inequities because they are often not registered and don't have legal status. This is especially a challenge when trying to assign unique IDs that can be connected to different existing formal or informal information systems.- Along the lines of COVID-19, I wonder if there is evidence of the rapidity of research design and deployment due to the urgency --- and at times at the expense of rigour and ethical considerations. Is there any evidence to suggest that? This seems to be implied in the framing but it is not elaborated on or justified.
---

	- Why was the Campbell Collaboration not included as a database? The Campbell Collaboration is an international social science research network that produces high quality, open and policy-relevant evidence syntheses, plain language summaries and policy briefs. - There is mention of 'Big Data' early on but nothing really specific about artificial intelligence, machine learning, etc. Was this within or outside the scope? There have been a number of guidelines, including from the WHO to support digital health reporting and guidelines that support inclusivity and redressing health inequities: https://www.nature.com/articles/s41746-020-00330-2 and https://www.bmj.com/content/352/bmj.i1174.abstract, among others. I would suggest referring to these at the outset. - the body of the paper seems very thin. There is no overview descriptive statistics of what has been found. It is unclear what contribution this paper will make in its current state. Despite the shortcomings of the paper in its current form, I believe this topic is very important and timely. If the changes can be made, it could provide worthwhile contributions to the body of evidence.
--	--

REVIEWER	Allison Crawford Centre for Addiction and Mental Health
REVIEW RETURNED	12-Jun-2022

GENERAL COMMENTS	Thank you for including me in the review of this protocol, “Researching the hard-to-reach: a scoping review protocol of digital health research in hidden, marginal, and excluded populations,” which is well-conceived and urgently needed. The manuscript is well-written and clear, and leverages appropriate frameworks and methods (Arskey and O’Malley; PRISMA-ScR; Peters et al.). I support publication of the protocol as-is, but have a few ideas for consideration, depending on the current stage of the review. 1. Grey literature search – some extra information about the methods for conducting the grey literature search would have been helpful. For example, there are also protocols for this – see, for example: Enticott J, Buck K, Shawyer F. Finding "hard to find" literature on hard to find groups: A novel technique to search grey literature on refugees and asylum seekers. Int J Methods Psychiatr Res. 2018 Mar;27(1):e1580. doi: 10.1002/mpr.1580. Epub 2017 Sep 4. 2. Digital health equity – in addition to the background on health equity supporting this work, there is an emergent literature on digital health equity (DHE). It may be useful for data abstraction and/or discussion to use a DHE framework. Here is an example (full transparency it is our framework): Crawford A, Serhal E. Digital Health Equity and COVID-19: The Innovation Curve Cannot Reinforce the Social Gradient of Health. J Med Internet Res. 2020 Jun 2;22(6):e19361. doi: 10.2196/19361. 3. Involving community members and people with living experience: One of the things that did strike me was the language, and I think it would be good to foreground this in some way in the presentation or discussion of results. For example, “hard to reach” “marginal” – one
---

	way to ensure critical perspectives, and especially to generate meaningful results, is to engage folks who are impacted by this research, and to pay them to be part of your working group at all stages. 4. Quality of included studies – The authors state, “the synthesis of data will be limited to a mapping of methods and categorization and less on the strength of each study due to the variance across the different fields of study and specializations.” I think it would be useful to try to identify aspects of the included studies that are associated with quality dimensions. Thank you again – I think the results of this review will be very impactful.
--	--

VERSION 1 – AUTHOR RESPONSE

Reviewer: 1

Ms. Chaitali Sinha, International Development Research Centre

Comments to the Author:

This paper is addressing a very timely and important issue. Please find below the concerns I have with it: Q1) Given the vast nature of the paper, encompassing several populations and the breadth of digital healthresearch methods and approaches, the details and analysis in the scoping review are not commensurate to what is needed for this type of contribution.

This is a protocol paper detailing our proposed scoping review. We agree with the reviewer that this is a timely and important issue, and therefore we aim to comprehensively review all available studies that focusing on the intersections between digital methods and hard-to-reach population inthe contexts of health research. Following the review and analysis, details on the results will be written for publication following a clear format (organized by different themes, for example). This will include an analysis of current stage of research, overall trends and potential future areas of research, taking stock of the current published literature at the intersection of hard-to-reach populations and digital methods.

Q2) Looking at the summary of strengths and limitations, I note that both the second and the third bullet are quite broad and rather unclear. For instance, it seems like the second bullet is about inequalities and inequities exacerbated by COVID-19, which are subsequently overlaid with other inequities in health andaccess to digital tools. But, this is a bit of a guess and the current wording makes me unsure what is beingcommunicated. General comment: there seems to be too many ideas being conveyed in this bullet.

Thank you for the suggestion. We agreed that the original statement in the second bullet was not clear. We now have revised the sentence completely, focusing on how this review may help informstrategies to address the issue of digital divide in health research among the populations.

The second bullet now reads,

“The rapid expansion in the use of digital health research risks further exacerbation of the digital divide among populations. This review will provide timely information on targeted strategies to promote and improve inclusivity of hard-to-reach populations in digital health research.”

Q3) The third bullet in this list also needs to be tightened up. Is the applicability wide because of the SDOH framing or because the inclusion criteria for the digital health research is not disease or population-specific?

Thank you for raising this. In response to this comment.

The third bullet now reads, *“The inclusion criteria for this scoping review is set to be broad, including any digital research on both the environmental and social determinants of health in hard-to-reach populations. This allows a thorough review of existing digital methods used across different sectors and disciplines that may have a wide applicability to hard-to-reach population research.”*

-Q4) I find some of the general statements like the first one in the paper a bit misleading. Currently it states: "Health equity is a common term in policy and research, although significant barriers remain towards achieving health for all." - I don't think the fact that a certain term is common in policy and research should translate into it being overcome. I would nuance this statement. Perhaps by saying that health equity is not applied consistently or systematically within or across populations and sub-populations.

Thank you for the suggestion. We have now deleted the first sentence, and added this *“Globally, health equity is not consistently or systematically applied within or across populations and sub-populations”.*

Q5) For the framing of the paper, it might be useful to state that marginalized, hidden and underserved populations are difficult to serve and redress health inequities because they are often not registered and don't have legal status. This is especially a challenge when trying to assign unique IDs that can be connected to different existing formal or informal information systems.

We agree with this statement. The lack of legal status is indeed one of the many barriers that these marginalized people face.

For the framing of the paper, we now write in the first paragraph of the Introduction:

“Populations residing in informal settlements, for example, face many socioeconomic, environmental and legal barriers that prohibit their access to health service. For example, they may lack formal government recognition, therefore their health status and needs are not tracked sufficiently by official systems. This poor data presents a formidable challenge for both researchers and policymakers to redress health inequities among these populations.”

However, please kindly note that the issue of registration and legal status probably only refers to specific marginalized populations – particularly residents of informal settlements, migrants and homeless populations. Disabled, LGBTQ+ and minority populations included in the review would not fall under this statement. It is therefore the recommendation of the authors not to provide any additional rationale on the

specific research barriers across different populations but consider this in the data extraction framework in order to understand how certain methods may be applied to different populations groups and/or barriers.

Q6) Along the lines of COVID-19, I wonder if there is evidence of the rapidity of research design and deployment due to the urgency --- and at times at the expense of rigour and ethical considerations. Is there any evidence to suggest that? This seems to be implied in the framing but it is not elaborated on or justified.

This comment was very helpful. Following a search there is documentation regarding the quality issues present COVID-19 research (commentary). A line has been added in the background of the paper and an additional reference to this point, with the link below.

<https://www.nature.com/articles/s41467-021-21220-5>

Added text: *“The speed and breadth of COVID-19 research was impressive, yet commentaries like those of Jung raise concerns regarding the adherence to research quality and standards in terms of study design, data quality and analytical methods. The latest guidance in the area of digital health such as the WHO Digital Health Guidelines and the Digital Health Equity Framework are opportunities to assess the quality of digital research.”*

-Q7) Why was the Campbell Collaboration not included as a database? The Campbell Collaboration is an international social science research network that produces high quality, open and policy-relevant evidence syntheses, plain language summaries and policy briefs.

The Campbell Collaboration was now included as a database as per the grey literature search strategy.

The revised text now reads;

-Q8) There is mention of 'Big Data' early on but nothing really specific about artificial intelligence, machine learning, etc. Was this within or outside the scope?

Research using 'Big Data', artificial intelligence, machine learning techniques will be within the scope of this review if they are in the context of health research in 'hard-to-reach' population. We have now updated the texts in Table 4 (Inclusion and Exclusion Criteria) about the digital application.

There have been a number of guidelines, including from the WHO to support digital health reporting and guidelines that support inclusivity and redressing health inequities: <https://www.nature.com/articles/s41746-020-00330-2>

The nature reference was not included as it related to digital intervention, however, reviews and commentaries will be part of the grey literature search and considered for the analysis and in the development of the data extraction framework.

and <https://www.bmj.com/content/352/bmj.i1174.abstract>, among others. I would suggest referring to these at the outset.

This article will be considered for the analysis for the development of the data extraction framework to ensure that the review captures the latest standards related to mobile technology.

Q9) the body of the paper seems very thin. There is no overview descriptive statistics of what has been found. It is unclear what contribution this paper will make in its current state.

Please kindly note that this is a protocol paper on the scoping review, rather than the scoping review itself. We have followed the journal's requirements in formatting this protocol paper and have refrained from sharing any preliminary results at this stage. As the reviewer already pointed out, this is a vast review potentially with many useful information for future works, it is therefore important to have this protocol paper reviewed by experts and later published to update researchers in the field about this particular research activity. The full scoping review will be submitted for publication in due course (November 2022).

Despite the shortcomings of the paper in its current form, I believe this topic is very important and timely.

If the changes can be made, it could provide worthwhile contributions to the body of evidence.

Reviewer: 2

Dr. Allison Crawford, Centre for Addiction and Mental Health

Comments to the Author:

Thank you for including me in the review of this protocol, "Researching the hard-to-reach: a scoping review protocol of digital health research in hidden, marginal, and excluded populations," which is well-conceived and urgently needed. The manuscript is well-written and clear, and leverages appropriate frameworks and methods (Arskey and O'Malley; PRISMA-ScR; Peters et al.).

I support publication of the protocol as-is, but have a few ideas for consideration, depending on the current stage of the review.

1. Grey literature search – some extra information about the methods for conducting the grey literature search would have been helpful. For example, there are also protocols for this – see, for example:

Enticott J, Buck K, Shawyer F. Finding "hard to find" literature on hard to find groups: A novel technique to search grey literature on refugees and asylum seekers. *Int J Methods Psychiatr Res.* 2018 Mar;27(1):e1580. doi: 10.1002/mpr.1580. Epub 2017 Sep 4.

This article was helpful (as well as the references) in ensuring that the strategy and scope of the grey literature search is understood. It is now documented in the search strategy Annex for reference.

In the texts, it now reads:

"While we primarily focus on peer-reviewed studies for this scoping review, we will also search grey literature, technical reports, and policy briefs produced by key agencies

following a novel approach asset out by Enticott et al (e.g. the World Health Organisation, UNICEF, the Campbell Collaboration.

These reports, guidance, and sector best practice will complement our primary evidence mapping basedon individual studies, to ensure the most complete view of evidence is achieved. Grey literature will be conducted following a novel approach as set out by Enticott et al. The search strategy is available in the supplementary file.”

We also reference the Digital Health Equity Framework in the introduction.

2. Digital health equity – in addition to the background on health equity supporting this work, there is an emergent literature on digital health equity (DHE). It may be useful for data abstraction and/or discussion to use a DHE framework. Here is an example (full transparency it is our framework):

Crawford A, Serhal E. Digital Health Equity and COVID-19: The Innovation Curve Cannot Reinforce the Social Gradient of Health. J Med Internet Res. 2020 Jun 2;22(6):e19361. doi: 10.2196/19361.

This article was added to the bibliography and referenced in the introduction.

3. Involving community members and people with living experience: One of the things that did strike me was the language, and I think it would be good to foreground this in some way in the presentation or discussion of results. For example, “hard to reach” “marginal” – one way to ensure critical perspectives, and especially to generate meaningful results, is to engage folks who are impacted by this research, and to pay them to be part of your working group at all stages.

Engagement of the community is now included in the data extraction framework to understand trends and how common co-design, community oversight etc. are used in digital research on vulnerable populations.

1. Quality of included studies – The authors state, “the synthesis of data will be limited to a mapping of methods and categorization and less on the strength of each study due to the variance across the different fields of study and specializations.” I think it would be useful to try to identify aspects of the included studies that are associated with quality dimensions.

Thank you for the suggestion. We agree that quality assessment of the reviewed studies are important. In the texts, it now reads:

“Quality of the reviewed studies will be assessed using checklists informed from Joanna Briggs Institute (JBI) (<https://jbi.global/critical-appraisal-tools>) for different types of studies.”

Thank you again – I think the results of this review will be very impactful.

VERSION 2 – REVIEW

REVIEWER	Chaitali Sinha International Development Research Centre, Global Health
-----------------	--

REVIEW RETURNED	31-Jul-2022
GENERAL COMMENTS	I have read the responses from the authors and am satisfied with this revised version of the manuscript. I feel it is ready to be published.